IFT-UAM/CSIC-25-063
June 16$^{th}$, 2025

# On the generalized Komar charge
# of Kaluza–Klein theories and higher-form symmetries

*Gabriele Barbagallo,*[1,a] *José Luis V. Cerdeira*[2,b] *Carmen Gómez-Fayrén,*[1,c]

*Patrick Meessen*[3,4d] *and Tomás Ortín*[1,e]

[1]*Instituto de Física Teórica UAM/CSIC*
*C/ Nicolás Cabrera, 13–15, C.U. Cantoblanco, E-28049 Madrid, Spain*

[2]*Instituto de Física Corpuscular (IFIC), University of Valencia-CSIC,*
*Parc Científic UV, C/ Catedrático José Beltrán 2, E-46980 Paterna, Spain*

[3]*HEP Theory Group, Departamento de Física, Universidad de Oviedo*
*Calle Leopoldo Calvo Sotelo 18, E-33007 Oviedo, Spain*

[4]*Instituto Universitario de Ciencias y Tecnologías Espaciales de Asturias (ICTEA)*
*Calle de la Independencia, 13, E-33004 Oviedo, Spain*

## Abstract

The generalized Komar $(d-2)$-form charge can be modified by the addition of any other on-shell closed (*conserved*) $(d-2)$-form charge. We show that, with Kaluza–Klein boundary conditions, one has to add a charge related to the higher-form symmetry identified in Ref. [10] to the naive Komar charge of pure 5-dimensional gravity to obtain a conserved charge charge whose integral at spatial infinity gives the mass. The new term also contains the contribution of the Kaluza–Klein monopole charge leading to electric-magnetic duality invariance.

---

[a]Email: `gabriele.barbagallo[at]estudiante.uam.es`
[b]Email: `jose.verez-fraguela[at]estudiante.uam.es`
[c]Email: `carmen.gomez-fayren[at]estudiante.uam.es`
[d]Email: `meessenpatrick[at]uniovi.es`
[e]Email: `Tomas.Ortin[at]csic.es`

# 1  Introduction

The impact and influence of Kaluza's discovery that some components of the 5-dimensional metric behaved as the 4-dimensional metric and Maxwell fields [1] complemented with Klein's realization that, if the 5[th] dimension is compact, the 4-dimensional charge and mass of a massless and uncharged particle moving in 5 dimensions are proportional to its momentum in that compact direction, inversely proportional to its size and quantized [2] in the subsequent development of Theoretical Physics cannot be overstated.[1] The main difference (and a very important one) between the gauge theories on which the Standard Model is based and Kaluza-Klein theories is that the extra dimensions of the former are not spacetime dimensions along which particles and gravity can propagate as in the second. As a matter of fact, the first non-Abelian Yang–Mills type theory was constructed by Pauli using Kaluza–Klein (KK) compactification in a space with $SU(2)$ symmetry [4] and the use of this mechanism in Supergravity and Superstring theories has played a fundamental role in the search for a unified theory of all interactions [5–7].

The KK paradigm "Physics in the uncompactified (*lower*) dimensions is just a manifestation of Physics in the total spacetime manifold (*higher dimensions*)" means that we should be able to derive no matter what lower-dimensional results working directly in higher dimensions. In this article we are concerned with Noether–Wald and generalized Komar charges and their use to study the thermodynamics of stringy black holes, finding their Smarr formulas and deriving the first law with all their charges and chemical potentials. These black holes are classical solutions of lower-dimensional Supergravity theories most of which can be derived by KK dimensional reduction from an 11- or 10-dimensional Supergravity[2] and it follows that we should be able to derive those Smarr formulas and first laws directly in 10 or 11 dimensions.

In order to achieve this goal

1. We need a higher-dimensional interpretation of all the conserved charges carried by the lower-dimensional black-hole solutions. This requires a complete dictionary between the local and global symmetries of the higher- and lower-dimensional theories.

2. We need a higher-dimensional interpretation of the associated chemical potentials.

3. We need a good understanding of the relations between the black-hole's higher- and lower-dimensional geometries. In particular, we need to know the relation between their event or Killing horizons.

4. With those interpretations we should be able to derive the Smarr formulas and first laws directly from the Noether–Wald and generalized Komar $(d-2)$-form

---

[1]See Ref. [3] for a review.
[2]For a recent review with many references see, for instance, Ref. [8].

charges ($d = 10, 11$).

In Refs. [9, 10] we have made progress in some of the points of this program using pure 5-dimensional Einstein gravity compactified on a circle. In Ref. [9] we studied the 5-dimensional geometry of the Killing horizons of static electrically-charged 4-dimensional black holes, finding that they are only stationary due to the uplifting to 5 dimensions of the timelike 4-dimensional Killing vector, which is a linear combination of the latter with the Killing vector generating translations along the compact direction. We also found that there is frame-dragging along the compact direction and that the velocity of the horizon in that direction is the 4-dimensional electrostatic potential. We also studied how the 5-dimensional Noether–Wald and Komar charges associated to the uplifted timelike Killing vector give rise to the 4-dimensional Smarr formula and 1st law. The additional terms proportional to the the Killing vector that generates translations along the compact direction give contributions proportional to the momentum in that direction which, as Klein found out, is seen as electric charge in 4 dimensions. However, we could not find the term proportional to the 4-dimensional magnetic charge in the 5-dimensional 1st law and the related term in the 4-dimensional Smarr formula only appeared after performing a rather *ad hoc* trick.

On the other hand, in Ref. [10] we found that the origin of the global symmetry of the 4-dimensional theory is a higher-form-type global symmetry which is only present in the 5-dimensional theory with the topologically non-trivial KK boundary conditions (one compact spatial dimension).

In this article we are going to show the relation between the higher-form symmetry found in Ref. [10] and the trick that allowed us to recover the full electric-magnetic duality-invariant Smarr formula in 4 dimensions in Ref. [9]. We are going to show that this trick is actually necessary if we want to have a generalized Komar charge for the Killing vector that generates time translations whose integral at infinity gives the mass.[3]

It is not difficult to see why the 5-dimensional Komar charge does not give the mass: the 4-dimensional metric and the 4-dimensional components of the 5-dimensional metric are related by a conformal transformation with conformal factor $k_\infty/k$ where $k$ is the KK scalar. The Komar integral at infinity simply picks the coefficient of $1/r$ in the the expansion of the $tt$-component of the metric

$$g_{tt} \sim 1 - \frac{2M}{r}. \tag{1.1}$$

However, if, at infinity

$$k \sim k_\infty \left(1 + \frac{\Sigma}{r}\right), \tag{1.2}$$

---

[3]It is important to clarify from the onset that in this context (5-dimensional gravity with KK boundary conditions) by mass of a solution we mean the mass of the 4-dimensional (dimensionally-reduced) solution in the 4-dimensional Einstein frame. While there are intrinsically 5-dimensional definitions of the mass in this context [11, 12], we believe that the above is the most natural notion.

where $\Sigma$ is the scalar charge, the expansion of the *tt*-component of the 5-dimensional metric will be

$$(k_\infty/k)g_{tt} \sim 1 - \frac{2M+\Sigma}{r}. \qquad (1.3)$$

The naive Komar integral will give a combination of the mass and the scalar charge and we have to modify it in a consistent way if we want it to give just the mass $M$.

As we are going to see, the trick used in Ref. [9] amounts to the addition of an on-shell closed (*conserved*) $(d-2)$-form charge to the naive generalized Komar charge (which is also on-shell closed, by construction) as explained in Ref. [13]. This addition does not change the value of the Smarr formula, since one adds and subtracts the same quantity,[4] but, when we only use the generalized Komar charge to compute the mass or other gravitational conserved charges integrating at infinity, the additional term does contribute to the result in a non-trivial way. In our case we will use this freedom to eliminate unwanted contributions.

We are going to start by reviewing the setup we are going to work with (pure 5-dimensional gravity with Kaluza–Klein boundary conditions) and the results of Refs. [9, 10] that we are going to use.

## 2   The setup

In this paper we are going to use the conventions of Ref. [19], with a mostly minus signature, the Vielbein $e^a = e^a{}_\mu dx^\mu$ as gravitational field and differential-form notation.[5] Furthermore, we use hats to denote the 5-dimensional quantities. In this language, the 5-dimensional Einstein–Hilbert action has the form

$$S[\hat{e}] = \frac{1}{16\pi G_N^{(5)}} \int \tfrac{1}{3!}\hat{\varepsilon}_{\hat{c}\hat{d}\hat{e}\hat{a}\hat{b}}\hat{e}^{\hat{c}} \wedge \hat{e}^{\hat{d}} \wedge \hat{e}^{\hat{e}} \wedge \hat{R}^{\hat{a}\hat{b}}(\hat{\omega}) \equiv \int \hat{\mathbf{L}}, \qquad (2.3)$$

where $G_N^{(5)}$ is the 5-dimensional Newton constant. We assume KK boundary conditions for the gravitational field: it admits a spatial Killing vector $k$ with closed orbits with no fixed points. We partially break the invariance under 5-dimensional general coordinate

---

[4]The Smarr formula can be obtained by integrating the generalized Komar charge over spatial infinity and over the horizon and equating the two results [14–18]. The on-shell closedness of the generalized Komar charge and the fact that these two $(d-2)$-dimensional surfaces are the only two pieces of the boundary of some hypersurface imply this equality.

[5]The Lorentz (spin) connection $\omega^{ab} = \omega_\mu{}^{ab}dx^\mu = -\omega^{ba}$ is assume to be the torsionless Levi-Civita connection $\omega^{ab} = \omega^{ab}(e)$

$$\mathcal{D}e^a \equiv de^a - \omega^a{}_b \wedge e^b = 0, \qquad (2.1)$$

and its curvature is defined as

$$R^{ab}(\omega) \equiv d\omega^{ab} - \omega^a{}_c \wedge \omega^{cb}. \qquad (2.2)$$

transformations (GCTs) by using as $5^{\text{th}}$ coordinate, $\hat{x}^4$, that we will denote by $z$, the coordinate adapted to $\hat{k}$. Thus $k = \partial_{\underline{z}}$ and all the components of the Vielbein are $z$-independent. Furthermore, since $z$ parametrizes the orbits of $\hat{k}$, $z \sim z + 2\pi\ell$ where $\ell$ is some length scale. This length scale must be distinguished from the length of the compact direction at a point $x$ of the 4-dimensional spacetime

$$L(x) = 2\pi R(x) = \int_0^{2\pi\ell} dz \sqrt{-\hat{g}_{\underline{zz}}(x)} = 2\pi\ell k(x)\,, \tag{2.4}$$

where $R(x)$ is the radius of the compact direction at the point $x$. The value of the radius at infinity is customarily denoted by $R$ and it is related to the value of the KK scalar at infinity $k_\infty$ by

$$R = \ell k_\infty\,. \tag{2.5}$$

Furthermore, we also partially break the invariance under 5-dimensional local Lorentz transformations with the following choice of Vielbein[6]

$$\hat{e}^a = e^a\,, \qquad\qquad \hat{e}_a = e_a - \imath_a A \partial_{\underline{z}}\,, \tag{2.6a}$$

$$\hat{e}^z = k(dz + A)\,, \qquad\qquad \hat{e}_z = k^{-1}\partial_{\underline{z}}\,, \tag{2.6b}$$

in which $e^a, A$ and $k$ will be identified with the 4-dimensional, KK-frame,[7] Vielbein, KK vector and KK scalar. Therefore, the 5-dimensional metric and 4-dimensional KK-frame fields are related by

$$ds^2_{(5)} = ds^2_{(4)} - k^2 (dz + A)^2\,, \tag{2.7a}$$

$$ds^2_{(4)} = g_{\mu\nu}dx^\mu dx^\nu\,. \tag{2.7b}$$

Under these assumptions and with these definitions, the action Eq. (2.3) and after integration over the coordinate $z$, can be rewritten in the form

$$S[e, A, k] = \frac{2\pi\ell}{16\pi G_N^{(5)}} \int \left\{ k \left[ - \star (e^a \wedge e^b) \wedge R_{ab} + \tfrac{1}{2}k^2 F \wedge \star F \right] + d\left[ 2 \star dk \right] \right\}\,. \tag{2.8}$$

---

[6]here $\imath_\xi$ indicates the interior product with the 4-dimensional vector $\xi$ and $\imath_a$ the interior product with $e_a$. Thus, $\imath_a A = e_a{}^\mu A_\mu$.

[7]In this conformal frame, the Einstein–Hilbert term in the action carries an additional scalar factor $k$. A local conformal rescaling of the metric and a global rescaling of the KK vector will be necessary to eliminate this factor and obtain the Einstein-frame action with no additional constants in it.

Finally, the rescalings

$$g_{\mu\nu} = (k/k_\infty)^{-1} g_{E\,\mu\nu}\,, \quad e^a{}_\mu = (k/k_\infty)^{-1/2} e_E{}^a{}_\mu\,, \quad A_\mu = k_\infty^{1/2} A_{E\,\mu}\,, \qquad (2.9)$$

bring us to the Einstein-frame action[8]

$$S[e_E, A_E, k] = \frac{1}{16\pi G_N^{(4)}} \int \left\{ -\star_E (e_E{}^a \wedge e_E{}^b) \wedge R_{E\,ab} + \tfrac{3}{2} d\log k \wedge \star_E d\log k \right.$$

$$\left. + \tfrac{1}{2} k^3 F_E \wedge \star F_E + d\left[ -\star_E d\log k \right] \right\}\,,$$
(2.10)

where the 4-dimensional Newton constant is given by

$$G_N^{(4)} = \frac{G_N^{(5)}}{2\pi R}\,. \qquad (2.11)$$

We can express the KK scalar in terms of an unconstrained[9] scalar field $k = e^{\phi/\sqrt{3}}$ with canonically-normalized kinetic term, but we will keep working with $k$ to keep the discussion as simple as possible.

## 2.1 The symmetries of 5-dimensional GR with KK boundary conditions

In view of the results of Ref. [10] it is convenient to revise the relation between the symmetries of the 5-dimensional theory and how they are related to the those of the 4-dimensional one.

The conventional wisdom is that the only symmetries (local or global) of GR are GCTs and, in our case, only those GCTs that preserve the KK boundary conditions. In the Vielbein formulation we must also include local Lorentz transformations that preserve those conditions and our choice Eq. (2.6). It turns out that GR with KK boundary conditions also admits higher-form-type global symmetries which had traditionally been incorrectly identified with a 1-parameter family of GCTs (see, for instance, Refs. [19, 9]).

Let us start with GCTs, which act on the 5-dimensional metric as

---

[8]The constant (modulus) $k_\infty$ must be introduced in the rescaling in order to ensure that the normalization of the 5- and 4-dimensional metrics at spatial infinity are the same. Sometimes this conformal frame is referred to as the *modified Einstein-frame* [20] to distinguish it from the one (standard in the earlier literature) in which this fact was not taken into account. In presence of matter fields, $k_\infty$ appears in the (modified) Einstein frame and one has to rescale some of the matter fields with it in order to make it disappear again. This is the origin of the definition of $A_E$. $k$ should not be rescaled if we do not want to modify its asymptotic value.

[9]$k$ has to be positive definite.

$$\delta_{\hat{\xi}}\hat{g}_{\hat{\mu}\hat{\nu}} = -\mathcal{L}_{\hat{\xi}}\hat{g}_{\hat{\mu}\hat{\nu}} = -\left(\hat{\xi}^{\hat{\rho}}\partial_{\hat{\rho}}\hat{g}_{\hat{\mu}\hat{\nu}} + 2\partial_{(\hat{\mu}}\hat{\xi}^{\hat{\rho}}\hat{g}_{\hat{\nu})\hat{\rho}}\right). \tag{2.12}$$

In order to preserve the $z$-independence of the metric, all the components except for $\hat{\xi}^z$ must be $z$-independent. Furthermore, the dependence $\hat{\xi}^z$ on $z$ must be, at most, linear. But vectors proportional to $z$ are not well defined because $z$ is multivalued. For this reason, the vector field $z\partial_{\underline{z}}$, which apparently generates constant rescalings of the 4-dimensional matter fields which are, actually a global symmetry of the 4-dimensional action Eq. (2.10), must be excluded. We will explain the 5-dimensional origin of that 4-dimensional symmetry in a moment.

Thus, taking into account the $z$-independence of the vector field $\hat{\xi}$, we can write the action of the 5-dimensional GCTs that respect the KK boundary conditions on the KK-frame 4-dimensional fields as

$$\delta_{\hat{\xi}}k = -\hat{\xi}^{\rho}\partial_{\rho}k, \tag{2.13a}$$

$$\delta_{\hat{\xi}}A_\mu = -\left(\hat{\xi}^{\rho}\partial_{\rho}A_\mu + \partial_\mu\hat{\xi}^{\rho}A_\rho\right) - \partial_\mu\hat{\xi}^z, \tag{2.13b}$$

$$\delta_{\hat{\xi}}g_{\mu\nu} = -\left(\hat{\xi}^{\rho}\partial_{\rho}g_{\mu\nu} + 2\partial_{(\mu}\hat{\xi}^{\rho}g_{\nu)\rho}\right), \tag{2.13c}$$

which correspond to 4-dimensional GCTs generated by the 4-dimensional vector field $\xi$ with components $\xi^\mu(x) = \hat{\xi}^\mu(x)$ and gauge transformations of the KK vector fields $A$ generated by the 4-dimensional gauge parameter

$$\hat{\xi}^z(x) = -\chi(x), \tag{2.14}$$

which act in the standard fashion: Lie derivative of the fields and

$$\delta_\chi A = d\chi, \tag{2.15}$$

respectively. Notice that these gauge transformations act on the compact coordinate $z$:

$$\delta_\chi z = -\chi(x), \quad \Rightarrow \quad \delta_\chi dz = -d\chi. \tag{2.16}$$

The 5-dimensional local Lorentz group is broken by our choice of (upper-triangular) Vielbein basis down to just the 4-dimensional one.

Let us now go back to 5-dimensional origin of the global symmetry of the 4-dimensional action Eq. (2.10) whose transformations have the form

$$\delta_\epsilon k = -\tfrac{2}{3}\epsilon k, \qquad \delta_\epsilon A_E = \epsilon A_E. \tag{2.17}$$

In Ref. [10] it was shown that the global transformation

$$\delta_\epsilon^h \hat{g}_{\hat{\mu}\hat{\nu}} = -2\epsilon\mathfrak{h}_{(\hat{\mu}} \hat{k}_{\hat{\nu})} \, , \tag{2.18}$$

where $\mathfrak{h} = \mathfrak{h}_{\hat{\mu}} d\hat{x}^{\hat{\mu}}$ is a harmonic 1-form and $\hat{k}$ is the Killing vector of the theory rescales the Einstein–Hilbert action with KK boundary conditions

$$\delta_\epsilon^h \hat{S} = -\epsilon\, \iota_k \mathfrak{h} \, \hat{S} = -\epsilon\hat{S} \tag{2.19}$$

where we assumed the normalization $\iota_{\hat{k}}\mathfrak{h} = 1$. A harmonic form $\mathfrak{h}$ typically exists in spacetimes with a compact direction such as those satisfying KK boundary conditions. Locally, and up to the total derivative of a $z$-independent function, $\mathfrak{h}$ can be written as $dz$. The addition of a total derivative to $\mathfrak{h}$ is fully equivalent to the transformation Eq. (2.16) and the gauge transformation Eq. (2.15). On the other hand, if $\mathfrak{h}$ is exact, then this transformation is equivalent to a GCT.

There is a second, independent, transformation that rescales the action

$$\delta_\epsilon^s \hat{g}_{\hat{\mu}\hat{\nu}} = \tfrac{2}{3}\epsilon\hat{g}_{\hat{\mu}\hat{\nu}} \, , \quad \Rightarrow \quad \delta_\epsilon^s \hat{S} = +\epsilon\hat{S} \, , \tag{2.20}$$

and both transformations can be combined into a global symmetry of the Einstein–Hilbert action with KK boundary conditions[10] $\delta_\epsilon \equiv \delta_\epsilon^h + \delta_\epsilon^s$ whose effect on the 4-dimensional fields is precisely Eq. (2.17) [10].

In what follows we are going to need the transformations of the Vielbein and spin connection under the above transformations. They are

$$\delta_\epsilon^h \hat{e}^{\hat{a}} = -\epsilon\iota_{\hat{k}} e^{\hat{a}}\mathfrak{h} \, , \qquad \delta_\epsilon^h \hat{\omega}^{\hat{a}\hat{b}} = -\epsilon\hat{P}_{\hat{k}}{}^{\hat{a}\hat{b}}\mathfrak{h} \, , \qquad \delta_\epsilon^h \hat{R}^{\hat{a}\hat{b}} = \epsilon\iota_{\hat{k}}\hat{R}^{\hat{a}\hat{b}} \wedge \mathfrak{h} \, , \tag{2.21a}$$

$$\delta_\epsilon^s \hat{e}^{\hat{a}} = \tfrac{1}{3}\epsilon\hat{e}^{\hat{a}} \qquad \delta_\epsilon^s \hat{\omega}^{\hat{a}\hat{b}} = 0 \, , \qquad \delta_\epsilon^s \hat{R}^{\hat{a}\hat{b}} = 0 \, , \tag{2.21b}$$

where $\hat{P}_{\hat{k}}{}^{\hat{a}\hat{b}}$ is the *Lorentz momentum map* or *Killing bilinear*, defined by the *momentum map equation*[11]

$$\iota_{\hat{k}}\hat{R}^{\hat{a}\hat{b}} + \mathcal{D}\hat{P}_{\hat{k}}{}^{\hat{a}\hat{b}} = 0 \, , \tag{2.22}$$

which always admits the solution

$$\hat{P}_{\hat{k}\,\hat{a}\hat{b}} = \mathcal{D}_{\hat{a}}\hat{k}_{\hat{b}} \, . \tag{2.23}$$

In this language it is very easy to obtain the transformation of the Einstein–Hilbert action Eq. (2.3). Ignoring the normalization factor $(16\pi G_N^{(5)})$

---

[10] This symmetry can be extended to include the coupling to matter fields [21].

[11] We have used this equation together with the Palatini identity $\delta\hat{R}^{\hat{a}\hat{b}} = \mathcal{D}\delta\hat{\omega}^{\hat{a}\hat{b}}$ to find $\delta_\epsilon^h \hat{R}^{\hat{a}\hat{b}}$ above.

$$\delta_\epsilon^h \hat{S} = \int \frac{1}{3!}\hat{\varepsilon}_{\hat{c}\hat{d}\hat{e}\hat{a}\hat{b}} \left\{ 3\delta_\epsilon^h \hat{e}^{\hat{c}} \wedge \hat{e}^{\hat{d}} \wedge \hat{e}^{\hat{e}} \wedge \hat{R}^{\hat{a}\hat{b}} + \hat{e}^{\hat{c}} \wedge \hat{e}^{\hat{d}} \wedge \hat{e}^{\hat{e}} \wedge \delta_\epsilon^h \hat{R}^{\hat{a}\hat{b}} \right\}$$

$$= -\epsilon \int \frac{1}{3!}\hat{\varepsilon}_{\hat{c}\hat{d}\hat{e}\hat{a}\hat{b}} \left\{ 3\imath_{\hat{k}} \hat{e}^{\hat{c}} \hat{e}^{\hat{d}} \wedge \hat{e}^{\hat{e}} \wedge \hat{R}^{\hat{a}\hat{b}} - \hat{e}^{\hat{c}} \wedge \hat{e}^{\hat{d}} \wedge \hat{e}^{\hat{e}} \wedge \imath_{\hat{k}} \hat{R}^{\hat{a}\hat{b}} \right\} \wedge \mathfrak{h}$$

(2.24)

$$= -\epsilon \int \imath_{\hat{k}} \hat{\mathbf{L}} \wedge \mathfrak{h}$$

$$= -\epsilon \hat{S}.$$

On the other hand, we can use the generic variation of the action

$$\delta \hat{S} = \int \left\{ \hat{\mathbf{E}}_{\hat{a}} \wedge \delta \hat{e}^{\hat{a}} + d\hat{\boldsymbol{\Theta}}(\hat{e}, \delta\hat{e}) \right\},$$

(2.25)

where the Einstein equations and presymplectic potential are given by

$$\hat{\mathbf{E}}_{\hat{a}} = \imath_{\hat{a}}\hat{\star}(\hat{e}^{\hat{a}} \wedge \hat{e}^{\hat{b}}) \wedge \hat{R}_{\hat{a}\hat{b}},$$

(2.26a)

$$\boldsymbol{\Theta}(\hat{e}, \delta\hat{e}) = -\hat{\star}(\hat{e}^{\hat{a}} \wedge \hat{e}^{\hat{b}}) \wedge \delta\hat{\omega}_{\hat{a}\hat{b}},$$

(2.26b)

respectively, to find an alternative expression for $\delta_\epsilon^h S$ and arrive at the identity

$$-\epsilon\hat{\mathbf{L}} = \hat{\mathbf{E}}_{\hat{a}} \wedge \delta_\epsilon^h \hat{e}^{\hat{a}} + d\hat{\boldsymbol{\Theta}}(\hat{e}, \delta_\epsilon^h\hat{e}),$$

(2.27)

which, gives as an expression for the on-shell Lagrangian as a total derivative[12]

$$\hat{\mathbf{L}} \doteq d\hat{\mathbf{J}}^h,$$

(2.28)

where we have defined the 4-form

$$\hat{\mathbf{J}}^h \equiv -\frac{1}{16\pi G_N^{(5)}} \left[ \hat{\star}(\hat{e}^{\hat{a}} \wedge \hat{e}^{\hat{b}})\hat{P}_{\hat{k}\hat{a}\hat{b}} \right] \wedge \mathfrak{h}.$$

(2.29)

We may use this result as explained in Ref. [13] to determine the generalized Komar charge of the theory. However, we observe that the 3-form that multiplies $\mathfrak{h}$ is nothing but the Noether–Wald charge associated to the Killing vector $\hat{k}$ of 5-dimensional GR. In this case, the Noether–Wald charge is nothing but (minus) the on-shell closed Komar charge [22]

---

[12] We use $\doteq$ for identities which hold on-shell.

$$\hat{\mathbf{K}}[\hat{k}] = \frac{1}{16\pi G_N^{(5)}} \hat{\star}(\hat{e}^{\hat{a}} \wedge \hat{e}^{\hat{b}})\hat{P}_{\hat{k}\,\hat{a}\hat{b}}\,, \tag{2.30}$$

and we can write

$$\hat{\mathbf{J}}^h = -\hat{\mathbf{K}}[\hat{k}] \wedge \mathfrak{h}\,. \tag{2.31}$$

Thus, it follows that $\hat{\mathbf{J}}^h$ is actually closed (conserved) and that the Lagrangian vanishes identically on-shell (a well-known fact in pure GR).

We could have arrived at the same result using $\delta_\epsilon^s$ for which we find

$$\epsilon \hat{\mathbf{L}} = \hat{\mathbf{E}}_{\hat{a}} \wedge \delta_\epsilon^s \hat{e}^{\hat{a}}\,, \tag{2.32}$$

with no total derivative, owing to the invariance of the spin connection under this rescaling $\delta_\epsilon^s \hat{\omega}^{\hat{a}\hat{b}} = 0$.

Combining these two facts, we arrive at the conclusion that $\hat{\mathbf{J}}^h$ is actually minus the Noether current associated to the global symmetry $\delta_\epsilon \equiv \delta_\epsilon^h + \delta_\epsilon^s$, which explains its on-shell conservation.

## 2.2 The 5-dimensional geometry of 4-dimensional stationary KK black holes

According to the rigidity theorem [25,26], the 4-dimensional geometry of 4-dimensional, asymptotically flat, stationary black holes is characterized by the existence of a Killing vector $m$ which is timelike close to infinity and a spacelike Killing vector field $n$ that generates rotations around a given axis. The Killing vector $l \equiv m - \Omega_{\mathcal{H}} n$, where the constant $\Omega_{\mathcal{H}}$ is the angular velocity of the horizon, becomes null over the event horizon $\mathcal{H}$

$$l^\mu g_{\mu\nu} l^\nu \overset{\mathcal{H}}{=} 0\,, \tag{2.33}$$

and $\mathcal{H}$ is, thus, identified as a Killing horizon.[13]

If this 4-dimensional black hole is a solution of the 4-dimensional KK theory Eq. (2.10) that we have obtained by dimensional reduction of GR with KK boundary conditions, it is also a solution of this 5-dimensional theory. As shown in [9], the 5-dimensional, asymptotically-KK solution is also stationary ($m$ is still a Killing vector and it is timelike near infinity) and also has a Killing horizon which is essentially the local product (a fibration) of the compact dimension with the 4-dimensional horizon. The Killing vector that becomes null over the 5-dimensional horizon (that we can call *the uplift* of $l$) is, actually [9]

---

[13]Notice that the above equation takes the same form in the Einstein and the KK frame because conformal transformations preserve the lightcone.

$$\hat{l} \equiv l - \chi_l \hat{k} = m - \Omega_{\mathcal{H}} n - \chi_l \hat{k}. \tag{2.34}$$

According to the discussion in the previous section and Eq. (2.14), the fact that $\hat{l}^{\hat{z}} = -\chi_{\hat{l}}$ means that the GCT generated by the 4-dimensional Killing vector will leave invariant the 5-dimensional metric if it is supplemented by a *compensating* or *induced gauge transformation* with parameter $\chi_l$, whose value we can determine by solving the 5-dimensional Killing vector equation for $\hat{l}$, $\mathcal{L}_{\hat{l}} \hat{g}_{\hat{\mu}\hat{\nu}} = 0$, assuming that $\hat{g}_{\hat{\mu}\hat{\nu}}$ satisfies the KK boundary conditions and is $z$-independent. We find three independent conditions:

$$\mathcal{L}_l k = 0, \tag{2.35a}$$

$$\mathcal{L}_l A - d\chi_l = 0, \tag{2.35b}$$

$$\mathcal{L}_l g_{\mu\nu} = 0, \tag{2.35c}$$

The second condition Eq. (2.35b), which confirms the interpretation of $\chi_l$ as an induced gauge transformation, can be rewritten in the form of the *momentum map equation*

$$\imath_l F + dP_l = 0, \qquad P_l = -\chi_l + \imath_l A. \tag{2.36}$$

The symmetry condition $\mathcal{L}_l F = 0$, together with the Bianchi identity $dF$ guarantee the local existence of the *momentum map* $P_l$, which is determined by the above equation up to an additive constant.

Thus,

$$\chi_l = \imath_l A - P_l. \tag{2.37}$$

Finally, the condition that $\mathcal{H}$ must be a Killing horizon of $\hat{l}$

$$\hat{l}^{\hat{\mu}} \hat{g}_{\hat{\mu}\hat{\nu}} \hat{l}^{\hat{\nu}} = l^{\mu} g_{\mu\nu} l^{\nu} - k^2 (\imath_l A - \chi_l)^2 \overset{\mathcal{H}}{=} 0, \tag{2.38}$$

together with Eq. (2.37) imply that we must choose the additive constant in $P_l$ so that

$$P_l \overset{\mathcal{H}}{=} 0. \tag{2.39}$$

# 3 The 5-dimensional generalized Komar charge

Linear combinations of conserved charges with constant coefficients are conserved. In our case this means that, as discussed in Ref. [13], if there is another conserved 3-form charge, we can add it to the standard Komar charge Eq. (2.30) with an arbitrary coefficient that we must determine using some physical criterion.

There are no other 3-form charges in 5-dimensional pure GR with asymptotically-flat boundary conditions, but there is one with KK boundary conditions, that we can derive from the Noether 4-form charge associated to the higher-form symmetry identified in Ref. [10].

As a general rule, given a $(d-1)$-form current $\mathbf{J}$ which is conserved when evaluated over a solution of the theory with a spacetime symmetry generated by the vector $p$ (*i.e.* $\delta_p$ annihilates all the fields of the solution and, therefore, the current), it is always possible to derive from it a $(d-2)$-form charge $\mathbf{Q}_p$ which is also conserved under the same assumptions [23].

In the case at hands, we are interested in stationary KK black holes which, on top of $\hat{k}$, admit Killing vector $\hat{l}$ given in Eq. (2.34) and we can derive a 3-form charge $\hat{\mathbf{Q}}_{\hat{l}}^h$ from the 4-form $\mathbf{J}^h$ using its on-shell conservation and the assumption $\delta_{\hat{l}}\mathbf{J}^h = 0$.

How does $\delta_{\hat{l}}$ act on $\mathbf{J}^h$? Let us first consider how $-\mathcal{L}_{\hat{l}}$ acts on $\mathfrak{h}$:

$$-\mathcal{L}_{\hat{l}}\mathfrak{h} = -d\imath_{\hat{l}}\mathfrak{h} = d\chi_l\,, \tag{3.1}$$

which, upon use of Eq. (2.16), we can rewrite in the form

$$\delta_{\hat{l}}\mathfrak{h} = -\mathcal{L}_{\hat{l}}\mathfrak{h} + \delta_{\chi_l}\mathfrak{h} = 0\,. \tag{3.2}$$

On the other hand, $\delta_{\hat{l}}$ acts on $\hat{\mathbf{K}}[\hat{k}]$ as minus the Lie derivative and, furthermore, since $\hat{l}$ is a Killing vector and $\delta_{\hat{l}}$ annihilates the Vielbein, metric etc., $\delta_{\hat{l}}\hat{\mathbf{K}}[\hat{k}] = \hat{\mathbf{K}}[\delta_{\hat{l}}\hat{k}] = 0$ because $\delta_{\hat{l}}\hat{k} = -[\hat{l}, \hat{k}] = 0$.

Then,

$$0 = \delta_{\hat{l}}\hat{\mathbf{J}}^h = -(d\imath_{\hat{l}} + \imath_{\hat{l}}d)\hat{\mathbf{J}}^h - \hat{\mathbf{K}}[\hat{k}] \wedge \delta_{\chi_{\hat{l}}}\mathfrak{h} \doteq -d\left\{\imath_{\hat{l}}\hat{\mathbf{J}}^h + \chi_{\hat{l}}\hat{\mathbf{K}}[\hat{k}]\right\}, \tag{3.3}$$

and we can define the 3-form charge

$$\hat{\mathbf{Q}}_{\hat{l}}^h \equiv \imath_{\hat{l}}\hat{\mathbf{J}}^h + \chi_{\hat{l}}\hat{\mathbf{K}}[\hat{k}]\,, \tag{3.4}$$

which is conserved (closed) when the hypotheses are satisfied: on-shell for solutions admitting the Killing vector $\hat{l}$. Since in Ref. [10] it was shown that the 5-dimensional 4-form current $\hat{\mathbf{J}}^h$ is related to the 4-dimensional 3-form current associated to the global symmetry Eq. (2.17), we expect 3-form charge Eq. (3.4) to be related to the 2-form charge that one can derive from the 4-dimensional current. The integral of this charge at infinity gives the scalar charge of the black hole [24, 23].

There are no more conserved charges in the theory under consideration and, therefore, we are led to consider the 1-parameter family of generalized Komar charges

$$\hat{\mathbf{K}}_\alpha[\hat{l}] \equiv \frac{1}{16\pi G_N^{(5)}} \left\{ \hat{\mathbf{K}}[\hat{l}] - \alpha\imath_{\hat{l}} \left[ \hat{\mathbf{K}}[\hat{k}] \wedge \mathfrak{h} \right] + \alpha\chi_{\hat{l}}\hat{\mathbf{K}}[\hat{k}] \right\}$$

$$(3.5)$$

$$= \frac{1}{16\pi G_N^{(5)}} \left\{ \hat{\mathbf{K}}[\hat{l}] - \alpha\imath_{\hat{l}}\hat{\mathbf{K}}[\hat{k}] \wedge \mathfrak{h} \right\}$$

since, by assumption, $\imath_{\hat{k}}\mathfrak{h} = 1$ and, therefore, $\imath_{\hat{l}}\mathfrak{h} = -\chi_{\hat{l}}$. In view of the previous comments, we expect that its integral at infinity will remove the contribution of the scalar charge mentioned in the Introduction (see Eq. (1.3)) for some value of $\alpha$. It is worth stressing that, as explained in the introduction, $\alpha$ has no influence on the Smarr formula.

In order to determine the value of $\alpha$ we have to study the pullback of $\mathbf{K}_\alpha[\hat{l}]$ over hypersurfaces that include the compact direction. The only components of $\mathbf{K}_\alpha[\hat{l}]$ whose pullback does not identically vanish are those that contain $\mathfrak{h}$ as factor.

For any 5-dimensional Killing vector $\hat{p}$, the standard Komar charge Eq. (2.30) has the following decomposition:

$$\hat{\mathbf{K}}[\hat{p}] = \frac{1}{16\pi G_N^{(5)}} \left[ -k \star (e^a \wedge e^b) \hat{P}_{\hat{p}\,ab} \right] \wedge \mathfrak{h} + 2 \star e^a \hat{P}_{\hat{p}\,az} - \left[ k \star (e^a \wedge e^b) \hat{P}_{\hat{p}\,ab} \right] \wedge A, \quad (3.6)$$

and, therefore, the terms of $\mathbf{K}_\alpha[\hat{l}]$ that contribute to the pullback are

$$\hat{\mathbf{K}}_\alpha[\hat{l}]_\mathfrak{h} = \frac{1}{16\pi G_N^{(5)}} \left\{ -k \star (e^a \wedge e^b) \left( \hat{P}_{\hat{l}\,ab} + \alpha\chi_l\hat{P}_{\hat{k}\,ab} \right) \right.$$

$$(3.7)$$

$$\left. -\alpha\imath_{\hat{l}} \left[ 2 \star e^a \hat{P}_{\hat{k}\,az} - \left[ k \star (e^a \wedge e^b) \hat{P}_{\hat{k}\,ab} \right] \wedge A \right] \right\} \wedge \mathfrak{h}.$$

Using the relation between the 5-dimensional Killing vector $\hat{l}$, the 4-dimensional one $l$ and $\hat{k}$ Eq. (2.34) and the expression Eq. (2.37) for the parameter of the compensating gauge transformation, we find

$$\hat{P}_{\hat{k}\,ab} = -\tfrac{1}{2}k^2 F_{ab}, \qquad\qquad \hat{P}_{\hat{k}\,az} = -\partial_a k,$$

$$(3.8)$$

$$\hat{P}_{\hat{l}\,ab} = P_{l\,ab} - \tfrac{1}{2}P_l k^2 F_{ab}, \qquad \hat{P}_{\hat{l}\,az} = -\tfrac{1}{2}k^{-1}\partial_a \left( k^2 P_l \right),$$

and, replacing these values in Eq. (3.7) we get

$$\hat{\mathbf{K}}_\alpha[\hat{l}]_\hbar = \frac{1}{16\pi G_N^{(5)}} \left\{ -k \star d\mathbf{l} - 2\alpha \star (dk \wedge \mathbf{l}) + (1-\alpha)P_l k^3 \star F - \alpha \tilde{P}_l F \right\} \wedge \hbar$$

(3.9)

$$+ d\left( \alpha \tilde{P}_l A \wedge \hbar \right),$$

where we have denoted by $\mathbf{l}$ the 1-form dual to the 4-dimensional Killing vector $l$ and we have defined the *dual (magnetic) momentum map* $\tilde{P}_l$.[14] The components of this 1-form are computed lowering the vector index with the KK-frame metric. Thus, according to Eq. (2.9),

$$\mathbf{l} = l^\mu g_{\mu\nu} dx^\nu = (k/k_\infty)^{-1} l^\mu g_{E\,\mu\nu} dx^\nu = (k/k_\infty)^{-1} \mathbf{l}_E.$$

(3.12)

Defining

$$P_l \equiv k_\infty^{1/2} P_{E\,l}, \qquad\qquad\qquad \imath_l F_E = -dP_{E\,l},$$

(3.13a)

$$\tilde{P}_l \equiv k_\infty^{1/2} \tilde{P}_{E\,l}, \qquad\qquad\qquad \imath_l \left( k^3 \star_E F_E \right) = -dP_{E\,l},$$

(3.13b)

and, discarding the total derivative, we arrive at[15]

$$\hat{\mathbf{K}}_\alpha[\hat{l}]_\hbar = \frac{k_\infty}{16\pi G_N^{(5)}} \left\{ -\star_E d\mathbf{l}_E + (1-2\alpha) \star_E (d\log k \wedge \mathbf{l}_E) + (1-\alpha)P_{E\,l} k^3 \star_E F_E - \alpha \tilde{P}_{E\,l} F_E \right\} \wedge \hbar.$$

(3.14)

The first term in $\hat{\mathbf{K}}_\alpha[\hat{l}]_\hbar$ is the standard 4-dimensional Komar charge associated to the 4-dimensional Killing vector $l$. That charge is not closed and two additional terms similar to the third and the fourth but with different coefficients, have to be added to construct the on-shell closed generalized Komar charge of the KK theory Eq. (2.10). The second term, on the other hand, gives an unwanted additional contribution which is proportional to the scalar charge of the KK scalar $k$. The same value of $\alpha$ that kills that term ($\alpha = 1/2$) give the right coefficients of the third and fourth terms and, using the relation between the 4- and 5-dimensional Newton constants Eq. (2.11), we finally arrive at

---

[14]Since

$$d\left( k^3 \star F \right) \doteq 0, \quad \text{and} \quad \mathcal{L}_l \left( k^3 \star F \right) = 0,$$

(3.10)

by assumption, $\imath_l \left( k^3 \star F \right)$ is closed on-shell, and there exists a function $\tilde{P}_l$ satisfying the *dual momentum map equation*

$$\imath_l \left( k^3 \star F \right) \doteq -d\tilde{P}_l.$$

(3.11)

[15]Notice that, for 2-forms $G$, $\star G = \star_E G$.

$$\hat{\mathbf{K}}_{1/2}[\hat{l}]_{\mathfrak{h}} = \frac{k_\infty}{16\pi G_N^{(5)}} \left\{ -\star_E d\mathbf{l}_E + \tfrac{1}{2} P_{El} k^3 \star_E F_E - \tfrac{1}{2} \tilde{P}_{El} F_E \right\} \wedge \mathfrak{h}$$

(3.15)

$$= \mathbf{K}[l] \wedge \frac{\mathfrak{h}}{2\pi\ell},$$

which, integrated over the compact direction gives the 4-dimensional generalized Komar 2-form charge.

Given the normalization of the momentum map $P_{El}$ (we had to impose that it vanishes over the horizon), the integral at infinity will include a contribution $\sim \Phi_\infty Q$ where $\Phi_\infty$ is the value of the corotating electrostatic potential at infinity.[16] In general, that value will not vanish. However, we are free to add on-shell closed 2-form charges like $-\Phi_\infty k^3 \star F_E \wedge \mathfrak{h}$ to remove that contribution. This can be done directly in 5 dimensions, adding a term $\sim \Phi_\infty \hat{\mathbf{K}}[\hat{k}]$. In practice, this is equivalent to replacing $\hat{l}$ by $\hat{l}' \equiv \hat{l} - \Phi_\infty \hat{k}$ which has the same form as $\hat{l}$ but with $P_l$ now vanishing at infinity instead of the horizon.

# 4 Discussion

In this paper we have shown how to construct a generalized Komar charge in 5-dimensional pure gravity with KK boundary conditions whose integral at spatial infinity for the Killing vector that generates time translations gives the mass, understood as the 4-dimensional mass, removing the contribution of the 4-dimensional KK scalar charge.

The construction is based on the freedom that one has to construct conserved charges as linear combinations of other conserved charges with constant coefficients and to the existence of a higher-form-type global symmetry in GR with KK boundary conditions [10] out of which we can derive conserved charges in symmetric solutions using the recipe of [9].

Since this higher-form-type global symmetry can be extended to the coupling with matter, it is interesting to see how the results of this paper can be generalized to more complex situations. Work in this direction is well underway [21].

# Acknowledgments

The work of GB, CG-F, TO and JLVC has been supported in part by the MCI, AEI, FEDER (UE) grants PID2021-125700NB-C21 ("Gravity, Supergravity and Superstrings" (GRASS)) and IFT Centro de Excelencia Severo Ochoa CEX2020-001007-S. The work of PM has been supported by the MCI, AEI, FEDER (UE) grant PID2021-123021NB-I00.

---

[16]We are free to impose that $\tilde{P}_{El}$ vanishes at infinity or, at least, that the integral of $\tilde{P}_{El} F_E$ does [27].

The work of GB has been supported by the fellowship CEX2020-001007-S-20-5. The work of CG-F was supported by the MU grant FPU21/02222. The work of JLVC has been supported by the CSIC JAE-INTRO grant JAEINT-24-02806. TO wishes to thank M.M. Fernández for her permanent support.

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
