# Peer review of "On the generalized Komar charge of Kaluza-Klein theories and higher-form symmetries"

_SciPost Physics Core_

## Round 1 · Referee Report · Anonymous (Referee 1) · 2025-8-20

Report

The authors investigate the definition of mass in 5 dimensional gravity with Kaluza-Klein boundary conditions that gives the correct mass from the four dimensional perspective. The work is interesting and deserves to be published.

Recommendation

Publish (easily meets expectations and criteria for this Journal; among top 50%)

---

## Round 1 · Referee Report · Anonymous (Referee 2) · 2025-8-27

Strengths

The paper expands and clarifies previous results by some of the authors, presented in refs. [9,10].

Weaknesses

It may not be immediately obvious what the use of the five-dimensional counterpart of the four-dimensional charge is, although the motivation for studying the higher-dimensional origin of gravitational conserved charges is clearly explained in the introduction.

Report

The authors consider five-dimensional Einstein gravity on a Kaluza-Klein spacetime given by the fibration of a Killing direction $S^1$ over four-dimensional spacetime, and study the five-dimensional counterpart of the generalized Komar integral which gives gravitational conserved quantities of a solution in four dimensions. Interestingly, the construction of the five-dimensional integral uses a higher-form symmetry based on the existence of a closed non-exact one-form in the five-dimensional theory with Kaluza-Klein boundary conditions. The paper continues work initiated in [9, 10], which is conveniently reviewed in section 2. A generalization to matter-coupled gravity is announced.

The presentation is clear, and it provides an appropriate amount of details and references. In my opinion, the paper is suitable for publication in SciPost in its current form.

Requested changes

The main result eq. (3.15) gives the five-dimensional counterpart of the Komar current associated with the Killing vector (called $l$) generating a black hole event horizon. This gives rise to a conserved charge which is the combination of the four-dimensional mass and angular momentum. On the other hand, at different points in the paper (including the abstract) it is stated that the five-dimensional counterpart of the integral giving the mass is obtained. I suggest the author clarify how one can obtain the mass and the angular momentum separately from (3.15).

A minor point: the indices should be fixed in eq. (2.26a).

Recommendation

Publish (easily meets expectations and criteria for this Journal; among top 50%)

  • validity: -
  • significance: -
  • originality: -
  • clarity: -
  • formatting: -
  • grammar: -

Author:  Tomás Ortín  on 2025-09-22  [id 5846]

(in reply to Report 2 on 2025-08-27)

Dear editor,

in order to answer the referee's point we have added the following paragraph below (3.15):

The first term in the right-hand side of the above equation is the standard
Komar charge whose integral gives the gravitational conserved charge
associated to the Killing vector $l$. In this case, $l$ is given by the linear
combination in Eq.~(\ref{eq:lhatdef}) (just the first two terms) when $m$
generates time translations and $n$ rotations around one axis. Since the Komar
charge is linear, its integral will give a linear combination of the mass and
and angular momentum with the angular velocity $\Omega_{\mathcal{H}}$ as
coefficient.

We have also corrected the typo in (2.26a)

We attach the PDF file of the corrected manuscript.

Yours,

Attachment:

KOMARKK.pdf

---

## Editorial Decision

resubmitted